# A parametrically programmable delay line for microwave photons

Takuma Makihara [1] ✉, Nathan Lee[1], Yudan Guo[1], Wenyan Guan[1] & Amir Safavi-Naeini [1] ✉

Delay lines that store quantum information are crucial for advancing quantum repeaters and hardware efficient quantum computers. Traditionally, they are realized as extended systems that support wave propagation but provide limited control over the propagating fields. Here, we introduce a parametrically addressed delay line for microwave photons that provides a high level of control over the stored pulses. By parametrically driving a three-wave mixing circuit element that is weakly hybridized with an ensemble of resonators, we engineer a spectral response that simulates that of a physical delay line, while providing fast control over the delay line's properties. We demonstrate this novel degree of control by choosing which photon echo to emit, translating pulses in time, and even swapping two pulses, all with pulse energies on the order of a single photon. We also measure the noise added from our parametric interactions and find it is much less than one photon.

Delay lines that preserve quantum information have been proposed as a resource for universal fault-tolerant quantum computing[1,2]. These works propose hardware-efficient approaches to quantum computing where the emission from a single well-controlled qubit is captured and stored in a long delay line to be interacted with at a later time. At optical frequencies, fibers have been used in experiments for generating two-dimensional cluster states, which are a resource for universal quantum computation[3,4]. In these implementations continuous variable quantum states are generated by squeezing the light field. An important parameter in these systems is the inverse of the squeezing bandwidth, which approximately determines the temporal extent of a mode and, therefore[5], the amount of delay needed to store it. Temporal widths smaller than $10^{-12}$ s have been realized in recent experiments allowing tens of meters of fiber to store on the order of $10^4$ modes simultaneously. In contrast, the quantum emitters used in discrete variable quantum systems typically emit photons much more slowly and need correspondingly longer delays. At microwave frequencies, artificial atoms constructed from superconducting circuits have already been used to demonstrate quantum advantage for certain problems[6–8]. The timescale for the emission of a microwave photon from these circuits is around $10^{-6}$ s. This makes implementing a delay line

challenging. For example, a superconducting coplanar waveguide (CPW) on silicon would need to be $\simeq 120$ meters in length to provide enough delay to store a just single mode.

Several innovative approaches have been developed to circumvent traditional delay lines' substantial physical length requirements. The first of these employs slow-wave or slow-light structures, which effectively reduce the speed of electromagnetic wave propagation by using metamaterials[9] or resonator arrays[10,11]. This technique allows for a significant decrease in the physical size of the delay line while maintaining its function. The second approach involves using waves that are not electromagnetic. A classic example is acoustic waves, such as those used by early mercury delay lines[12], and more recent Surface Acoustic Wave (SAW) technologies[13] which have been applied for delaying quantum information[14]. These methods use waves with inherently slower propagation speeds, in this case acoustic, to achieve the desired delay in a reasonable amount of space. Lastly, approaches that use atoms or emitters based on Electromagnetically Induced Transparency (EIT)[15], or those using atomic frequency combs (AFCs)[16,17] have been developed in the last decades with a view towards quantum information. The latter AFC schemes are noteworthy as they emulate the response of a traditional delay line through light-matter interactions facilitated by pump fields. They realize what is, in effect, a virtual delay line – even if the system behaves as if a localized pulse is

---

[1]Department of Applied Physics, Stanford University, Stanford, California, USA. ✉e-mail: makihara@stanford.edu; safavi@stanford.edu

 1

propagating down a waveguide, in reality, the excitation is in the coherences of a large number of atoms and may be distributed in space and frequency in a manner that does not resemble a localized propagating pulse. The unique advantage here lies in the ability to alter the characteristics of the delay line simply by modifying the pump fields, offering a dynamic solution for implementing a more robust and versatile delay line.

In this study, we introduce a Parametrically Addressed Delay Line (PADL) as a versatile virtual delay line for microwave photons, leveraging pump fields that drive parametric processes to dynamically control the speed, direction, coupling strength, and connection points of the signal within the line. We implement this virtual delay line by parametrically distributing a data pulse that is launched at a lumped element readout mode into excitations in an ensemble of long-lived resonators. Controlling all of the delay line's properties translates to controlling the parametric drive frequencies, amplitudes, and phases. We show how a parametric delay line gives us more control than a physical waveguide by: (1) controlling the drive amplitudes to selectively choose the number of round trips that a wavepacket makes inside the delay line, (2) controlling the drive phases to translate the pulse in time, and (3) controlling the drive detunings to swap two wavepackets in time. A key question is how the delay line will perform for quantum pulses and whether the PADL's parametric nature leads to excess noise, such as through parasitic processes, to degrade performance. For this, we measure the added noise from the parametric drives by calibrating the gain of our measurement apparatus using our resonators as quantum microwave parametric oscillators (MPOs)[18] that operate as quantum-calibrated sources of microwave radiation. Specifically, we use the number of photons in our quantum MPO near the threshold as an in-situ noise power calibration device[19], and find that the added noise is much less than one photon per mode.

## Results

### Implementing a parametric delay line

The PADL works by parametrically distributing a data pulse that is launched at a readout mode (referred to as the buffer mode) into excitations of a collection of long-lived storage resonators. By controlling the parametric drives, we can engineer the buffer mode's

spectrum to emulate the characteristics of a reflective delay line. As a simple example, consider using PADL to emulate a physical delay line with a free spectral range (FSR) given by $\Omega$. This can be achieved with PADL by continuously parametrically coupling the buffer mode and the storage resonators such that the detuning $\Delta_k$ between the converted photons from the $k^{\text{th}}$ storage resonator and the buffer form a frequency comb with an FSR (peak spacing) $\Omega$ and such that the coupling strengths realize the desired loading. Figure 1a illustrates the relevant frequencies in this example. The storage resonator frequencies are labeled by $\omega_k$, and the buffer frequency is labeled by $\omega_b$. The parametric couplings are illustrated by dashed lines, and the parametrically converted storage photons are shown to have detunings $\Delta_k$ with an FSR given by $\Omega$. The storage resonator frequencies do not need to be precisely placed or tuned–we use the parametric drive frequencies to compensate for any disorder to realize the mode spacing $\Omega$[20]. Figure 1b illustrates this mode of operation by introducing an analogy to a physical delay line (illustrated as a ring supporting data pulses). For continuous parametric coupling, the delay line modes evolve by accruing a phase $\phi_k = \Delta_k t$ and rephase after a round-trip time $T_{\text{rt}} = 2\pi/\Omega$, such that the output pulse (drawn with a solid line) is simply the input pulse (drawn with a dashed line) delayed by $T_{\text{rt}}$.

However, the PADL provides far more exotic dynamics than a simple physical delay because we can parametrically program the delay line mode couplings, phases, and detunings. Crucially, these couplings are independently tunable; changing the $k^{\text{th}}$ parametric drive changes the coupling to the $k^{\text{th}}$ storage resonator. In Fig. 1c, we illustrate how we can effectively disconnect the virtual delay line from the input/output by shutting off the parametric drives, i.e., by setting their amplitudes $|\epsilon_k| = 0$. This prevents the rephased signal from being emitted into the environment and causes it to propagate around the virtual delay line for longer. Turning the drives back on reconnects the waveguide to the environment and causes the pulse to be re-emitted at the next round-trip time $NT_{\text{rt}}$ for some positive integer $N$. This mode of operation is analogous to fiber loop buffers which have been recently considered for storing quantum information[21]. Full control over the phases of the drives allows us arbitrary access to information stored in the delay line. In Fig. 1d, we illustrate how we continuously translate the pulse forwards or backward in time by an amount $\tau$ (modulo $T_{\text{rt}}$) by

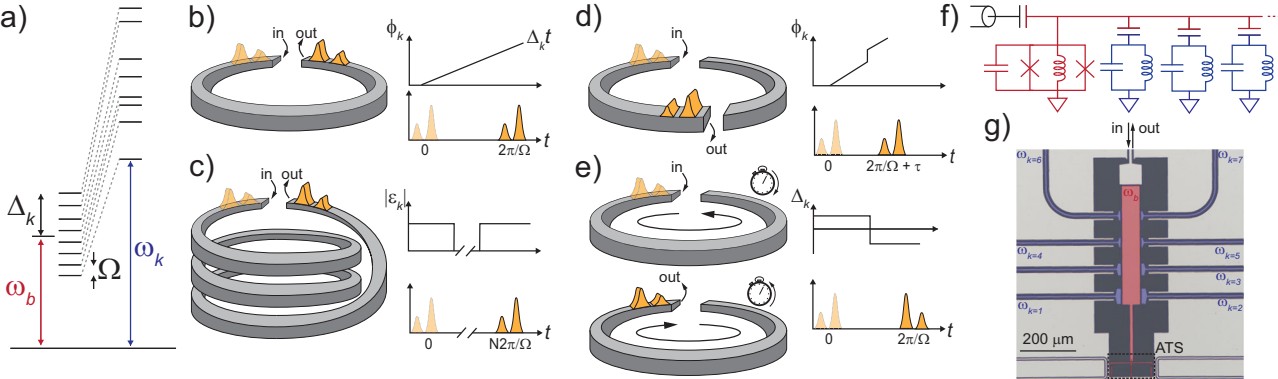

**Fig. 1 | Principles of the PADL and its implementation using an ATS. a** We implement a parametric delay line by parametrically coupling an ensemble of resonators (with frequencies $\omega_k$) to a readout mode (with frequency $\omega_b$, which we refer to as the buffer mode). The parametric drives are indicated by dashed lines. We engineer the buffer mode's spectrum into looking like a delay line with an FSR denoted by $\Omega$ by choosing the converted storage photons to have detunings $\Delta_k$ relative to $\omega_b$ and to form a frequency comb with an FSR given by $\Omega$. **b** Analogy between a physical waveguide delay line (illustrated as a ring supporting pulses) and the PADL under continuous parametric driving. In this system, the $k^{\text{th}}$ delay line mode evolves by accruing a phase $\phi_k = \Delta_k t$. The input pulses are illustrated by dashed lines, and the delayed pulses are illustrated by solid lines. **c** By turning off all

parametric drive amplitudes $|\epsilon_k|$ when a photon echo is about to rephase, one can prevent the signal from emitting into the environment and thereby selectively emit a later photon echo. **d** By instantaneously translating the $k^{\text{th}}$ delay line mode's phase by $\phi_k \to \phi_k + \Delta_k \tau$, one can translate the pulse by a time $\tau$ (modulo $2\pi/\Omega$). **e** By swapping the detunings of the delay line modes $\Delta_k \to -\Delta_k$ one effectively take $t \to -t$ in the phase accrued by the delay line modes and swap the order of two pulses. **f** Circuit diagram of our parametric delay line implementation. We weakly hybridized CPW resonators (illustrated by lumped LC resonators) with one ATS. **g** False-color optical micrograph of our device, where the CPW resonators are shown in blue and the lumped element buffer mode is shown in red. The ATS is indicated by a dashed box.

instantaneously translating each phase by $\phi_k \to \phi_k + \Delta_k \tau$. This is analogous to accessing the pulse at positions other than the open port in the virtual delay line. In Fig. 1e, we illustrate how controlling the parametric drive frequencies allows us to swap two pulses in time. Specifically, changing $\Delta_k \to -\Delta_k$ is equivalent to taking $t \to -t$ in terms of the phase that the virtual delay line modes accrue. This is analogous to switching the pulse propagation direction in the virtual delay line. While we focus on these four experiments, one can engineer more complicated pulse dynamics with the novel degree of control provided by a parametric delay line.

We implement the physical circuit on-chip by fabricating quarter-wavelength CPW resonators that are weakly capacitively coupled with a nonlinear resonant circuit known as an Asymmetrically Threaded SQUID (ATS)[22]. The ATS circuit is given by two nominally identical Josephson junctions forming a loop with an inductive shunt in the center of the loop. We use an array of Josephson junctions to form the inductive shunt. The circuit diagram is illustrated in Fig. 1f, where we illustrate the CPW resonators by their equivalent lumped element LC models. A false-color optical micrograph of the PADL device is shown in Fig. 1g, where the buffer mode is highlighted in red, and the CPW resonators are highlighted in blue.

The ATS provides the lumped element buffer mode that we use to readout our pulses, as well as the necessary nonlinearity to parametrically couple the different modes of our circuit. We choose to use an ATS for three reasons. Firstly, the ATS provides three-wave mixing as opposed to four-wave mixing – the native nonlinearity of Josephson junctions. Three-wave mixing allows us to parametrically swap between the ATS lumped element mode and the CPW modes while minimizing the adverse effects from four-wave-mixing interactions that are characteristic of multimode systems connected to junctions. Secondly, the ATS has an inductor, which provides an unconfined parabolic potential and therefore its lumped element mode can be strongly driven before it becomes nonlinear. Finally, we can use the three-wave mixing nonlinearity to implement a quantum MPO in the CPWs, which we can use for noise power calibration.

When the ATS is precisely biased at its "saddle-point," the energy associated with the phase drop across the junctions changes from the usual $\cos(\varphi)$ to $\sin(\varphi)$. This crucial modification shifts the dominant nonlinear term from $\varphi^4$ to $\varphi^3$. We use this altered nonlinearity to enable the three-wave mixing processes essential for the operations conducted here. The Hamiltonian at the saddle-point is

$$\hat{H} = \hbar\omega_b \hat{b}^\dagger \hat{b} + \sum_k \hbar\omega_k \hat{a}_k^\dagger \hat{a}_k \\ - 2E_J \epsilon_p(t) \sin\left(\varphi_b(\hat{b}+\hat{b}^\dagger) + \sum_k \varphi_k(\hat{a}_k+\hat{a}_k^\dagger)\right), \tag{1}$$

where $\varphi_b$ is the node flux zero-point fluctuation (ZPF) of the buffer mode across the ATS Josephson junctions with annihilation operator $\hat{b}$ and frequency $\omega_b$. Similarly $\varphi_k$ is the ZPF of the $k^{\text{th}}$ CPW resonator fundamental mode at the same circuit node with annihilation operator $\hat{a}_k$ and frequency $\omega_k$. $E_J$ is the individual junction energy in the SQUID, and $\epsilon_p(t)$ is a time-dependent parametric flux pump threading the SQUID. Note that we have assumed $|\epsilon_p(t)| \ll 1$.

We resonantly select specific interactions by driving the buffer mode while simultaneously flux pumping the SQUID. We focus here on a beamsplitter interaction between the buffer mode and the $k^{\text{th}}$ CPW's fundamental mode. We flux pump the SQUID at a single frequency $\omega_p$ and drive the buffer mode with multiple drives tones, as captured by the following driving Hamiltonian:

$$\hat{H}_{\text{drive}}/\hbar = \sum_k \left(\epsilon_k \hat{b} e^{i\omega_{d,k}t} + \text{h.c.}\right). \tag{2}$$

Here, $\omega_{d,k}$ is the frequency of a detuned drive on the buffer with field amplitude $|\epsilon_k|$. We choose the drive frequencies to satisfy

$$\omega_{d,k} = \omega_p - (\omega_b + \Delta_k) + \omega_k. \tag{3}$$

In the frame where both the CPW mode and the buffer are rotated out, our total Hamiltonian approximately becomes[23]:

$$\hat{H}/\hbar = \sum_k \Delta_k \hat{a}_k^\dagger \hat{a}_k + \sum_k g_k \hat{a}_k^\dagger \hat{b} + \text{h.c.} \tag{4}$$

where $\hbar g_k = E_J \epsilon_p \varphi_b^2 \varphi_a \beta_k$ is the parametric coupling strength between the buffer and the $k^{\text{th}}$ CPW's fundamental mode, and $\beta_k$ is the small displacement on $\hat{b}$ generated by the $k^{\text{th}}$ drive. By operating the buffer in the fast-cavity regime, i.e., $\kappa_{b,e} \gg g_k$, we can adiabatically eliminate it ($\kappa_{b,e}$ is the extrinsic loss rate of the buffer mode). We tune the drive amplitudes and frequencies so that the $g_k$ and harmonically placed $\Delta_k$ in the resulting effective Hamiltonian closely resemble that of a delay line. Importantly, by controlling the parametric drives' frequency, amplitude, and phase, we control the corresponding delay line mode's detuning, coupling, and phase. In principle, the flux pump frequency is arbitrary because the drive frequencies can be chosen to satisfy Eq. (3). In practice, one could be limited by the amount of microwave power that is available to drive the buffer.

The ATS parameters are chosen such that $\omega_b/2\pi = 5.0073$ GHz and $\omega_k/2\pi \simeq 6.91 - 7.46$ GHz (see Supplementary Note 1 for more details). The buffer is capacitively coupled to the environment at a rate $\kappa_{b,e}/2\pi = 3.95$ MHz, and the resonator intrinsic quality factors approximately range from $110 \times 10^3$ to $300 \times 10^3$. We estimate from finite element electromagnetic simulations (see Supplementary Note 1 for more details) that the buffer impedance and hybridization strengths with the CPWs leads to flux ZPF across the junction for each mode of roughly $\varphi_b = 0.336$ and $\varphi_k \simeq 0.018 - 0.023$.

## Parametric control of stored wavepackets

We first start with the simplest PADL experiment – implementing a response that mimics that of a reflectively terminated transmission line probed at the other end. For this, we tune and fix the parametric drives' amplitudes, phases, and detunings (as illustrated in Fig. 1b) for seven of the CPWs. For all the experiments in this work, we only parametrically couple seven of the eight CPW resonators to the buffer as we observed one of the CPW modes to have larger frequency fluctuations, which we attribute to a nearby two-level system (TLS) defect. We set the FSR to be $\Omega/2\pi = 500$ kHz so the delay line bandwidth closely matches the buffer mode's extrinsic coupling rate. The continuous wave (CW) flux pump tone is provided by a signal generator (Keysight E8257D PSG). The parametric drive intermediate frequency (IF) tones are all generated on a single arbitrary waveform generator (AWG) channel (Tektronix 5200) before being up-converted to drive the buffer mode. Before being up-converted, the AWG output is amplified by a room-temperature low-noise amplifier. All pulses sent into the delay line are played and demodulated using an Operator-X (OPX) from Quantum Machines Inc. (QM), and similarly, the pulses are up-converted and down-converted using an Octave from QM. The demodulated pulses are also digitized on an analog-to-digital converter (ADC) in the OPX (see Supplementary Note 2 for more details). We emphasize that the flux pump tone provides the magnetic flux $\epsilon_p(t)$ through the ATS loops and is fed to the device via a transmission line that is grounded near the ATS, whereas the parametric drives $\epsilon_k(t)$ are fed through the readout transmission line that is capacitively coupled to the buffer mode (see Supplementary Note 2 for more details).

We probe the PADL on reflection near the buffer mode frequency and clearly observe the parametrically coupled modes in the normalized reflection coefficient $S_{11}(\omega)$ (Fig. 2a). We tune the parameters to obtain an approximately linear phase response centered at 5.0321 GHz

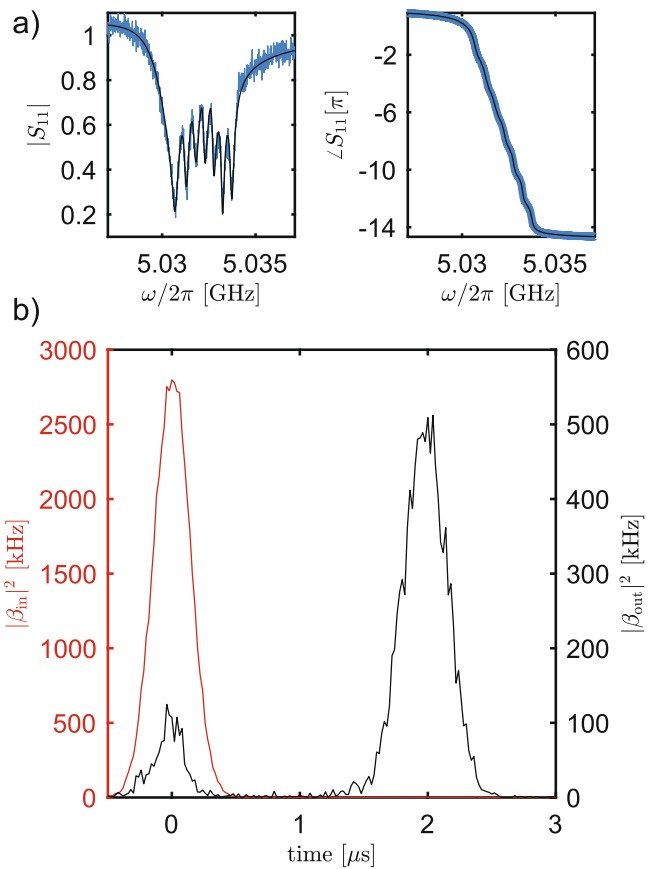

**Fig. 2 | A continuously coupled parametric delay line. a** The reflection coefficient of our buffer mode ($S_{11}(\omega)$) when seven CPW resonators are parametrically coupled to the mode. **b** Analog-to-digital converter traces of a pulse that is stored (black) and not stored (red) in our parametric delay line. The delayed pulse is delayed by approximately $T_{rt} = 2\,\mu s \simeq 2\pi/\Omega$. The time-domain traces are reported in units of photon flux (see Supplementary Note 6 for more details).

to emulate the $\omega T_{rt}$ phase response of a physical delay line. We measure the time-domain response by sending pulses at the PADL. Figure 2b shows the results of reflecting an attenuated Gaussian pulse ($\langle n \rangle \simeq 1$, with a temporal FWHM of 471 ns; see Supplementary Note 6 for more details on the calibration of the attenuation). The red pulse centered at $t = 0$ results from reflecting a pulse off of the device in the absence of pump and drives and with the buffer mode far-detuned so that the device acts as a mirror. The black pulse results from reflecting a pulse off the buffer mode while the device is emulating a delay line. We observe that the pulse is approximately delayed by $2\,\mu s \simeq T_{rt}$. The small reflected pulse at $t = 0$ in the black trace is due to the mismatched impedance between the environment and the parametric delay line, including any inevitable mismatches from device packaging (see Supplementary Note 5 for more details). The time-domain traces are reported in units of input photon flux $|\beta_{in}|^2$ and output photon flux $|\beta_{out}|^2$. These fluxes are related by the input-output boundary condition $\beta_{out} = \beta_{in} + \sqrt{\kappa_{b,e}}\beta$, where $\beta = \langle \hat{b} \rangle$.

By turning the parametric drive amplitudes $|\epsilon_k|$ off, we prevent a stored wavepacket from being emitted, causing the pulse to undertake another round trip in the delay line, as illustrated in Fig. 1c. Turning the amplitudes back on allows the next photon echo to be emitted. In Fig. 3a, we sweep the duration $\tau$ over which we turn off the parametric drive amplitude. Delay $\tau = 0$ corresponds to the drives being on continuously. We see that as $\tau$ exceeds $T_{rt}$, the first echo disappears, and the second echo at $2T_{rt}$ becomes more prominent. Similarly, as $\tau$ exceeds $2T_{rt}$, the second photon echo disappears, and the third photon echo at $3T_{rt}$ becomes more prominent. In Fig. 3b, we

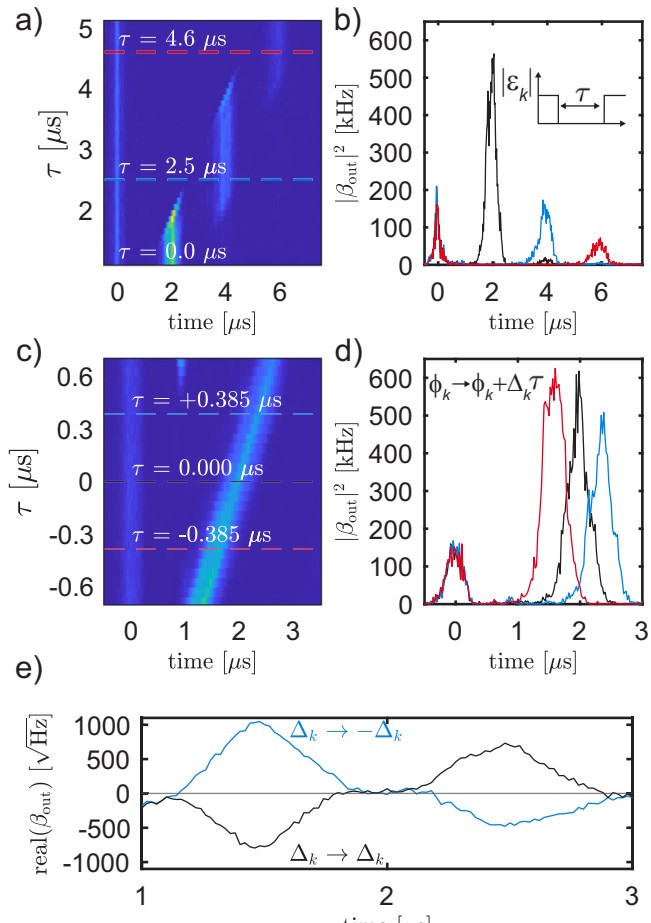

**Fig. 3 | Parametric control of the delayed pulse. a, b** By turning off the parametric drive amplitudes for a duration $\tau$, one can selectively emit later photon echoes. **a** A color map of ADC traces for $\tau$ swept from $0\,\mu s$ to $5\,\mu s$. **b** ADC traces for $\tau = 0.0\,\mu s$ (black), $2.5\,\mu s$ (blue) and $4.6\,\mu s$ (red). **c, d** By instantaneously translating the phase of the $k^{th}$ parametric drive by an amount $\Delta_k\tau$, one can translate the pulse in time by $\tau$ (modulo $2\pi/\Omega$). **c** A color map of ADC traces for $\tau$ swept from $-0.7\,\mu s$ to $+0.7\,\mu s$. **d** ADC traces for $\tau = -0.385\,\mu s$ (red), $0.000\,\mu s$ (black), and $+0.385\,\mu s$ (blue). **e** By instantaneously swapping the detunings $\Delta_k \rightarrow -\Delta_k$ in the parametric delay line, one can swap two pulses in time.

show the ADC data of traces taken with $\tau = 0\,\mu s$ (black), $2.5\,\mu s$ (blue), and $4.6\,\mu s$ (red).

We fine-tune the photon emission time by modifying the parametric drive phases to modify $\phi_k$. By changing these phases, we continuously translate the pulse in time, as illustrated in Fig. 1d. Specifically, by translating $\phi_k \rightarrow \phi_k + \Delta_k\tau$, we can translate our pulse by a time $\tau$ modulo $T_{rt}$. In Fig. 3c, we sweep the duration $\tau$ by which we translate our pulse, where $\tau = 0$ corresponds continuous parametric driving without phase modification. We see that as we sweep $\tau$, the time at which the pulse re-emits translates linearly. In Fig. 3d, we show the ADC data of traces taken with $\tau = 0.000\,\mu s$ (black), $-0.385\,\mu s$ (red), and $+0.385\,\mu s$ (blue).

Finally, we show how controlling the detunings of the parametrically converted CPW photons $\Delta_k$ allows us to swap two pulses in time, as illustrated in Fig. 1e. Swapping the detunings $\Delta_k \rightarrow -\Delta_k$, is analogous to performing time reversal $t \rightarrow -t$ in the phase accrued by the delay line modes, causing the first stored pulse to be emitted after the second pulse. In Fig. 3e, we show two complex ADC traces: one where the detunings are swapped (blue) and one where they are not (black). In this experiment, we send two pulses with FWHM = 283 ns and a separation of 1000 ns. We make the first pulse have a $\pi$ phase shift relative to the second pulse and plot one quadrature of the

measured ADC data to clearly show that the two pulses are swapped in order when the detunings are swapped. The black trace in Fig. 3e plots the case where the detunings are not swapped, corresponding to the case of continuous drives. The blue trace in Fig. 3e plots the case where the detunings are swapped, corresponding to the case where we take $t \to -t$. We clearly see that relative to the black trace, the pulses have been swapped. In Supplementary Note 5, we provide additional numerical simulations considering two pulses with different relative amplitudes and phases.

## Fidelity and added noise

An important figure of merit in quantum memories is the fidelity of the state being read out compared to that which was stored. Ideally, the output pulse should be identical in amplitude and phase to the input pulse. In reality, distortions and loss induced by storage in the PADL and thermal noise reduce the fidelity. We will show that the latter is negligible in our system and first focus on distortion and loss. Given two pulses that are characterized by temporal mode operators $\hat{A}_1 = \int dt f(t)\hat{a}(t)$ and $\hat{A}_2 = \int dt g(t)\hat{a}(t)$, we define the fidelity as $F = |\int dt f^*(t) g(t)|^2$, where $f(t)$ is the temporal mode profile for the input pulse and $g(t)$ is the temporal mode profile for the delayed pulse. Here, we focus on the fidelity of the pulse that is delayed in the presence of continuous parametric drives (such as the black pulse in Fig. 2b) such that $g(t) \simeq f(t - T_{rt})$. We calculate these mode profiles by normalizing the detected field measured on the ADC. The input mode is always normalized such that $\int dt |f(t)|^2 = 1$ corresponding to unit detection probability. To isolate the impact of distortions on the pulse while it is stored in the PADL, we first normalize the delayed pulse such that $\int dt |g(t)|^2 = 1$. We also exclusively focus on the pulse centered at $2\,\mu s$ and neglect contributions from the small reflected pulse at $t = 0$, which can be caused by impedance mismatches in the device packaging. With this normalization for $g(t)$, which ignores losses and effectively compares the shape of the delayed temporal mode to the input temporal mode, we measure $F = 0.95$. Time-domain simulations of the semi-classical equations of motion for $\hat{a}(t)$ and $\hat{b}(t)$ derived from Eq. (4) show that fidelity can be further improved by optimizing the parametric delay line parameters and input pulse bandwidth (see Supplementary Note 5 more details). To include the effect of losses, we normalize $g(t)$ by the same factor that we normalize $f(t)$, resulting in $\int dt |g(t)|^2 \leq 1$. In this case, we find a $F = 0.21$ that includes both loss and distortion, showing that the infidelity is primarily due to losses in the storage resonators. The infidelity from storage resonator loss is neither limited by the loss of the worst resonator nor the sum of all resonator losses. Given that the input pulse is parametrically distributed into the collection of storage resonators, the input pulse's frequency components near $\Delta_k$ are predominantly attenuated by the $k^{th}$ resonator's loss rate.

In addition to loss and distortion, an important concern is whether the parametric driving of the PADL leads to excess noise being added to the microwave field. We estimate the added noise by measuring the power spectral density of the field emitted at the CPW mode frequencies when the drives are on. The key challenge is to calibrate the gain and loss in the readout signal path with sufficient precision to be able to infer the microwave fluctuations at the device from the field detected outside the fridge. Previously, this has been accomplished by using in-situ calibrated sources, such as shot noise tunnel junctions[24] or qubits[25]. In both these cases, the physics of the source is sufficiently well-understood to provide a signal with a well-defined photon flux or noise power without requiring a component-by-component accounting of gain and loss. Here, we use the device itself, operated as a parametric oscillator, as such a quantum-calibrated source. In a classical model of a parametric oscillator there is a discontinuity at the oscillation threshold. This discontinuity is smoothed away by the fluctuations of the electromagnetic field in a more accurate quantum

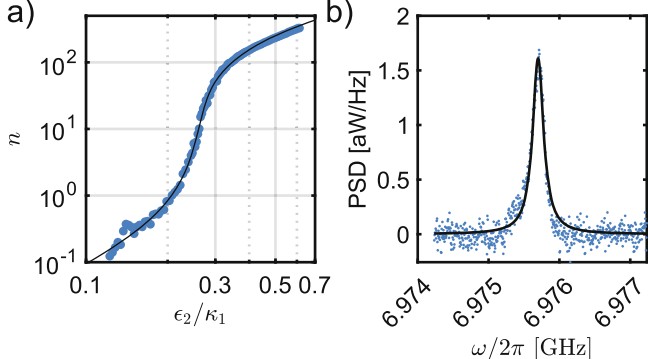

**Fig. 4 | Calibration of added noise. a** A plot of the number photons $n$ vs. normalized two-photon drive strength $\epsilon_2/\kappa_1$ as one MPO crosses threshold. Both axes are log scale. **b** Spectrum of the same resonator when the pump and drives used to generate a parametric delay line are applied on the device.

model of a parametric oscillator. The dependence of number of intracavity photons $n$ vs. drive power obtains a characteristic shape[19,26] which we use to infer the number of photons and calibrate the gain of our amplification chain (see Supplementary Note 6 for more details).

We implement the MPO on the same device by pumping the ATS at $\omega_p = 2\omega_k - \omega_b$ to resonantly select a 2-photon swapping term given by $\hat{a}_k^2 \hat{b}^\dagger + \hat{a}_k^{2\dagger}\hat{b}$. After adiabatically eliminating the fast-decaying buffer mode and applying a resonant drive, the effective dynamics of the $k^{th}$ CPW mode are governed by a Lindblad master equation $\dot{\rho} = -i[\hat{H}, \hat{\rho}] + \mathcal{D}[\hat{L}_1]\hat{\rho} + \mathcal{D}[\hat{L}_2]\hat{\rho}$, with the Hamiltonian and loss operators given by

$$\hat{H} = i\epsilon_2 \hat{a}^2 + \text{h.c.}, \quad \hat{L}_2 = \sqrt{\kappa_2}\hat{a}^2, \quad \text{and} \quad \hat{L}_1 = \sqrt{\kappa_1}\hat{a}. \tag{5}$$

In these equations, $\epsilon_2$ is the two-photon drive strength, $\kappa_2$ is the two-photon loss rate, and $\kappa_1$ is the single-photon loss rate.

In Fig. 4a, we show how the steady-state CPW intracavity photon number $n$ changes with the amplitude of the normalized two-photon drive when we operate one of the CPWs (the resonator at 6.975562 GHz) as a quantum MPO. In the following experiments, we directly readout the individual CPWs through the readout port, which is possible due to weak parasitic capacitances. We measure the integrated power spectral density (PSD), which is proportional to $n$, on an RF spectrum analyzer and plot the result versus the amplitude of the driving field sent to the buffer mode. A smooth transition is clearly visible and the data agrees closely with the quantum model (solid black line) of the MPO (see Eq. (5)) with three fitting parameters: (1) the proportionality constant between $\epsilon_2/\kappa_1$ and the driving field at the instrument, (2) the proportionality constant between $n$ and the integrated PSD at the instrument, which is related to the gain, and (3) $\kappa_2/\kappa_1$.

With the gain calibrated, we can determine the number of added noise photons. In Fig. 4b, we show the spectrum of the same CPW resonator as in Fig. 4a but now with the parametric drives and pump for the delay line experiments turned on. By fitting this spectrum to a Lorentzian (solid black line) to extract the spectral area and by using our calibrated gain, we conclude that 0.11 noise photons are added to this mode when it is operated as the PADL. We find that the added noise in all modes is always less than $\simeq 0.15$ photons for the reported drive intensities. The added noise is probably due to heating from our strong parametric flux pump. This pump power is comparable to previous quantum-limited parametric amplifiers[27,28] and is likely comparable to previous ATS-based devices with higher quality-factor resonators demonstrating cat states[22,29]. Therefore, we do not believe this small added noise will hinder future quantum applications for the PADL.

## Discussion

Unlike a waveguide delay line, the PADL gives us complete control over the detunings, phases, and coupling rates of the delay line modes. Furthermore, delays that are comparable to a several kilometer long waveguide can be achieved for microwave photons in a small footprint. Unlike catch-and-release methods that require a single resonant mode and precise mode matching via dynamic and precisely timed control of cavity parameters, PADL offers far greater flexibility. By using a three-wave mixing circuit element we sidestep issues due to parasitic processes that arise more frequently in four-wave mixing schemes. In lieu of performing process tomography on encoded qubits[30], which would enable the evaluation of the delay as a quantum process, we characterize the bosonic channel by measuring the number of photons of noise that are added into our delay line as well as the overlap between the input and output wavepackets. We demonstrate dynamic programmability by selecting the emission of later photon echoes, translating pulses continuously in time, and swapping two pulses stored in the emulated delay line.

Finally we stress that more intricate control can be engineered, given that the PADL is fully programmable and that integration with qubits is a possibility. Furthermore, by integrating qubits on the same chip as the PADL, one can address any impedance mismatches that arise from packaging. Nonetheless, practical use as a quantum memory will require much higher fidelities. One simple way to improve fidelity is to use larger bandwidth pulses and shorter delays such that the delay is much less than the CPW resonator lifetime. Ultimately, improving the resonator lifetime is important for improving fidelity. In future work, integration with recently developed high-Q Tantalum CPW resonators[31] or microwave cavities[32,33] coupled to ATSs[19,22,29] may open the route to long programmable delays in quantum processors with much higher fidelity.

## Methods

### Device fabrication

Our device is patterned in aluminum on 525 $\mu$m thick high-resistivity silicon ($\rho > 10$ k$\Omega \cdot$cm). The sample is first solvent cleaned in acetone and isopropyl alcohol, followed by the following four-mask process:

1. Etched alignment marks: Alignment marks are patterned with photolithography (Heidelberg MLA150 direct-writer) and etched into the sample using XeF$_2$. The sample is then cleaned in baths of piranha (3:1 H$_2$SO$_4$:H$_2$O$_2$) and buffered oxide etchant.
2. Circuit patterning: Ground planes, CPWs, flux lines, and the ATS island are patterned with photolithography, followed by a gentle oxygen plasma. Aluminum is deposited in an electron beam evaporator (Plassys) and lifted off in N-Methyl-2-pyrrolidone (NMP).
3. Junction patterning: The ATS itself (including the SQUID loop and the superinductor) are patterned by electron beam lithography (Raith Voyager), followed by a gentle oxygen plasma. Aluminum is deposited at an angle of 62°, followed by oxidation at 50 Torr for 10 minutes, followed by aluminum deposition at an angle of 0°. Liftoff is performed in NMP. The junction-array inductor is formed from 21 junctions using a Dolan-bridge method, whereas the single junctions in the SQUID are formed using a T-style process[34].
4. Bandaging: We use a bandage mask to ensure a superconducting connection between the previous two masks. The bandages are patterned with electron beam lithography and overlap both masks. Prior to aluminum deposition (at 0°), we ion-mill in-situ to clear away any oxide, thus ensuring a superconducting connection. Liftoff is performed in NMP.

### Circuit Analysis

The ATS consists of a SQUID (with individual junction energies $E_J$) that is threaded by an inductor (with energy $E_{L_b}$). In terms of the node flux operator $\hat{\varphi}$ of the ATS node, the potential from the inductor and

SQUID can be written as:

$$U = \frac{1}{2}E_{L_b}\hat{\varphi}^2 - 2E_J\cos(\varphi_\Sigma)\cos(\hat{\varphi} + \varphi_\Delta) \quad (6)$$

where:

$$\varphi_\Sigma = (\varphi_{\text{ext},1} + \varphi_{\text{ext},2})/2 \quad (7)$$

$$\varphi_\Delta = (\varphi_{\text{ext},1} - \varphi_{\text{ext},2})/2 \quad (8)$$

and where $\varphi_{\text{ext},1}$ and $\varphi_{\text{ext},2}$ are the external magnetic fluxes threading the left and right loops formed by the inductor and a Josephson junction[19,22,29].

When the device is flux-biased to $\varphi_\Sigma = \varphi_\Delta = \pi/2$ and a small RF modulation $\epsilon_p(t)$ is applied to $\varphi_\Sigma$, the potential (to first order in $\epsilon_p$) becomes:

$$U = \frac{1}{2}E_{L_b}\hat{\varphi}^2 - 2E_J\epsilon_p(t)\sin(\hat{\varphi}) \quad (9)$$

Since the ATS is capacitively coupled to other modes, the node flux operator $\hat{\varphi}$ representing the flux across the junction can be written in terms of the normal modes of the linear circuit as:

$$\hat{\varphi} = \varphi_b(\hat{b} + \hat{b}^\dagger) + \sum_k \varphi_k(\hat{a}_k + \hat{a}_k^\dagger) \quad (10)$$

where $\varphi_b$ is the node flux ZPF of the "buffer-like" normal mode, and $\varphi_k$ is the node flux ZPF of the "CPW-like" normal mode of the $k^{\text{th}}$ CPW mode[35]. The full Hamiltonian is then:

$$\hat{H} = \hbar\omega_b\hat{b}^\dagger\hat{b} + \sum_k \hbar\omega_k\hat{a}_k^\dagger\hat{a}_k \\ - 2E_J\epsilon_p(t)\sin\left(\varphi_b(\hat{b} + \hat{b}^\dagger) + \sum_k \varphi_k(\hat{a}_k + \hat{a}_k^\dagger)\right) \quad (11)$$

as discussed above.

Microwave Parametric Amplification: When $\epsilon_p(t)$ is pumped at a frequency $\omega_p = 2\omega_k - \omega_b$, we need to look for terms in the Hamiltonian that can resonate with this frequency. They are of the form[23]:

$$\varphi_b\varphi_k^2\left(\hat{a}_k^2\hat{b}^\dagger + \hat{a}_k^{\dagger 2}\hat{b}\right).$$

These terms represent processes where two photons from the $k^{\text{th}}$ CPW mode are either absorbed or emitted in combination with the emission or absorption of a photon from the buffer-like mode.

Beamsplitter operation: Alternatively, when $\epsilon_p(t)$ is pumped at a frequency $\omega_p = \omega_b + \omega_{d,k} - \omega_k$, and a second drive at frequency $\omega_{d,k}$ is applied to the buffer, we will implement a beam splitter interaction between the buffer-like mode and the $k^{\text{th}}$ CPW mode. In this setup, the interaction Hamiltonian can be simplified to terms that resonate with the combined effect of both drives. The resonant terms in this case are[23]:

$$\varphi_b^2\varphi_k(\hat{b}^\dagger\hat{a}_k + \text{h.c.})$$

## Data availability

The source data for Fig. 2, Fig. 3, and Fig. 4 generated in this study are provided with the paper and its supplementary information files. Source data are provided with this paper.

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

## Acknowledgements

The authors thank O. Hitchcock, R. Gruenke, M. Maksymowych, T. Rajab-zadeh, Z. Wang, W. Jiang, F. Mayor, S. Malik, S. Gyger, and E. A. Wollack for useful discussions and assistance with fabrication. The authors thank K. Villegas and T. Gish from QM for their assistance with the Octave and OPX. The authors wish to thank the following sources of financial support for this work: NTT Research, the National Science Foundation CAREER award No. ECCS-1941826, the Q-NEXT DOE NQI Center, and Amazon Web Services Inc. This material is based upon work supported by the Air Force Office of Scientific Research and the Office of Naval Research under award number FA9550-23-1-0338. Any opinions, findings, and conclusions or recommendations expressed in this material are those of the author(s) and do not necessarily reflect the views of the United States Air Force or the Office of Naval Research. Device fabrication was performed at the Stanford Nano Shared Facilities (SNSF) and the Stanford Nanofabri-cation Facility (SNF), supported by the NSF award ECCS-2026822. T.M. acknowledges support from the National Science Foundation Graduate Research Fellowship Program (grant no. DGE-1656518).

## Author contributions

A.S.N. conceived of the project. T.M. designed, fabricated, and measured the device with assistance from N.L., Y.G., and W.G. The manuscript was written by T.M. and A.S.N with comments from all other authors.

## Competing interests

The authors declare no competing interests.

## Additional information

**Peer review information** : *Nature Communications* thanks Ofer Naaman and the other, anonymous, reviewer(s) for their contribution to the peer review of this work. A peer review file is available.

