## [Peer Review File · Nature Communications]

A parametrically programmable delay line for microwave photonsREVIEWER COMMENTS

Reviewer #1 (Remarks to the Author):

The authors introduce the concept of a Parametrically Addressable Delay Line (PADL) as a parametrically programmable delay line. They demonstrate its main features and functionalities. Remarkably they show translating pulses in time and swapping of two pulses. I believe that such a PADL will be of great interest to the community working on the realization of quantum computing with superconducting circuits. I think that this work should be granted publication in Nature Communications, if the authors reasonably address my concerns, reported below.

1. The first paragraph in the introduction should be expanded, and focus more on why delay lines can be instrumental for realizing quantum computing. There are only two references given and I would expect to see more. Also, references on the realization of a quantum computer based on superconducting circuits are missing.
2. I have some hard time understanding what is the real fidelity of the device. In the “fidelity and added noise” section, the authors first say that they measure fidelity of 95%, but when they then normalize $g(t)$ by the same factor as $f(t)$, the fidelity drops to 21%. So, what is the true device fidelity? My concern is then, would a device with such a low-fidelity be useful? Is there any threshold value that one should aim at achieving? How can one improve it, without using the ultra-high Q cavities mentioned by the authors? How much would the fidelity improve using state-of-the-art CPW resonators with higher Q?
3. In the conclusion, the authors suddenly talk about performing “process tomography on encoded qubits”. This comes out of the blue and no reference or any explanations are given. should put references when talking about performing

I have also some minor comments:

4. The “Results” section starts with a subtitle reading “Implementing a parametric delay line with ATS”. Until this point there is no mention of what an ATS is in the text, so ATS should be removed from the subtitle.
5. Acronym for FSR is not introduced to the readers
6. In the caption of Fig 2, the authors talk about ADC traces. Again, ADC needs to be defined

somewhere if it is used. Also, if the traces are reported in terms of photon flux β_{in} and β_{out} , these quantity needs to be defined in the main text rather than in the supplementary information. Same comments apply to the caption of Figure 3. “ADC data” is also mentioned elsewhere in the text.

7. In the supplementary, Figures should be called with a different label. Maybe Fig #S, to distinguish them from figures in the main text Also the acronym ADC is used without any introduction

Reviewer #2 (Remarks to the Author):

The manuscript describes a method for emulating a compact tunable delay line using parametric techniques. The chip uses 7 on-chip CPW cavities capacitively coupled to a buffer resonant mode that is modulated by a flux applied to an asymmetric tunable SQUID inductance. By applying appropriate parametric drives it is possible to control and manipulate microwave pulses that can reflect off of the buffer mode that is coupled to an input/output feedline. The Authors show the ability to delay the pulse over short times, delay for longer times by storing it in the resonators by turning off the parametric drives for storage time, rapidly stepwise advance or translate the phase, and finally, swap the order of pulses by reversing the evolution in time.

I find this work novel and a practical tool for use in future quantum information systems.

These features have not been achieved before in a very compact and convenient design that uses the same circuit resources already employed in existing superconducting circuits for quantum computing. I think this work will be of significance to the field and related fields and that these techniques will find use in helping develop future quantum information processors and communication channels. I believe that work presented supports the conclusions and claims, but I do have further comments and questions below that should be addressed. I believe there is enough detail provided in the methods for the work to be reproduced, but again see my comments below.

I think the work could be published in Nature Communications in principle after edits based on the comments below:

1) Fig. 1: It'd be good to show a zoom in of the junctions in the ATS. At the scale shown, it's

hard to tell what the JJs look like or how many there are.

2) Fig. 1: The cartoon pictures are also confusing to me. I believe the red pulse (same color as the buffer resonator?) is not delayed and the blue pulse (same color as the resonator modes that get parametrically coupled) is delayed, however they are close together in the graph with time and far apart on the cartoon of the waveguide? What are the two colors for, can you specify? Why is there two red and two blue pulses in the graphs? Is this meant to represent something at a beginning time and then at an ending time? It would be clearer to show one red and blue pulse delayed in one plot, then have another plot at a later time to show the delay between the two pulses with respect to each other at a later time -i.e., are they farther apart, closer together, or did they swap positions. The waveguides look clear but the blue and red pulses on them seem too far apart on these. Also, I think the little curved arrows are trying to say the pulse went into the delay line or came off it? Maybe describe this in the caption and/or label in and out?

3) I see 8 CPW's coupled to the buffer, but only 7 modes listed in the paper?

4) What are the ghost like streaks in Fig. 3a) at constant time at 2, 4, and 6 us, extending from τ of 0 to 2 us, 2 to 4 us, and 4 to probably 6 us, respectively. Are they a concern for high fidelity operation?

5) It seems that the Fidelity integral should be a function of τ . As seen in Fig. 3c) the amplitude of the shifted pulse decreases in amplitude with longer τ . How will this effect performance and usefulness of the delay line? If the Fidelity is only maximum at T_{rt} , what can be done to improve performance over a larger range of τ ?

6) In Fig. 3e), although the pulse appear reversed in time, the amplitudes suggest they were just flipped over. Would a simulation removing loss and reflections show improvement for this behavior?

7) Shouldn't $\kappa_b = \kappa_{b,e} + \kappa_{b,i}$ be used in equation (1) of SM?

8) I assume $\kappa_k = \kappa_{k,e} + \kappa_{k,i}$, so how did you separate the values for $\kappa_{k,e}$ & $\kappa_{k,i}$ for the fits in Table II?

9) Figure 5 in SM was not there. I looked at the Arxiv version Fig. 9.

10) The end of the paper would be improved by discussing what improvements must be made for practical use. SM Note 4 models the system and tries to improve the fidelity. Can these tests point to a direction for improvement? How can losses be improved in the CPW modes? Could they be lumped-elements instead? How can the mode spacings be compensated if not equal (as stated in the intro) or made to be more equal to ensure performance? Could tweaks be made to operational frequencies and pulses to improve existing performance? How can the imperfections in impedance mismatches be solved? I think these questions and their answers must be included in the paper to show this device's viability.

Reviewer #3 (Remarks to the Author):

The authors demonstrate a device, built with an asymmetrically threaded squid (ATS) parametrically connecting several superconducting CPW resonators, which the authors can operate as an effective delay line. Operation relies on the ATS mediating frequency conversion between an input mode (called the “buffer mode”) and the ensemble of resonators, with carefully tuned (detuning, phase, and amplitude) parametric pumps. The authors show microsecond-scale programmable delay of an input pulse, storage of pulses, and temporal swapping of pulse sequences. Programmability is achieved by dynamically controlling the parametric pump detunings. The authors additionally calibrate their device parameters, specifically the added noise, with separate experiments operating the device in amplifier/oscillator mode near threshold.

This is a really cool experiment with a novel device. While the application space is not very well articulated, I think this type of device (and indeed the device concept) could become an important tool in superconducting quantum information processing, perhaps in architectures based on cavity bosonic states. I therefore would support publication in

Nature Communications, but I would like the authors to address some issues detailed below, mainly with the presentation of the work.

1. Analogy of delay line. I think the way that the analogy is introduced, and is used throughout the paper perhaps is going a bit too far to be useful. The introduction, in combination with the description associated with Fig. 1 (a-e), planted in my head a concept of the device that is actually the wrong one, and not until middle of page 4 I was able to actually understand what the experiment is actually doing and what the device topology is. My (hopefully now correct) understanding of the experiment is that an input pulse is spread by parametric conversion into a collection of resonant storage modes. Control of the detunings of the pumps can arrange it so that the pulse only interferes constructively back at the buffer I/O mode at particular times, hence it is effectively delayed. I would strongly recommend that the authors present a high level description of what's actually going on here (perhaps along the lines of my attempted explanation above), before driving the delay-line analogy. I also find Fig. 1 (a-d) not particularly helpful, and if the authors could come up with a more intuitive visualization that refrains from evoking a physical extended line or fiber topology for the experiment, that would improve the accessibility of the paper.

2. Pump tuneup and setting the detunings. An important experimental bit of information is how to choose the correct detunings for all the pumps, and I find that information missing.

- What expression would we use to target these detunings, and how this is done procedurally in the lab.
- Are pump phases important? Do the pumps have to be coherent, and to what accuracy? What would be the effect of fixed phase offsets on the pumps? What would be the effect of relative phase drifts?

3. Why ATS? A key component of the device is the ATS, which serves as part of the buffer mode, and is also providing the parametric coupling when pumped. I find the explanation of why they use an ATS not entirely convincing. The 3 reasons given on Page 2 would suggest that a DC SQUID could be used instead (3 wave mixing, inductive reactance, can use in MPO mode). Is there a specific property about the nonlinearity of the ATS that makes it advantageous in this application, over a flux pumped DC SQUID?

4. Are there 7 or 8 CPW resonators?

- Fig. 1g shows what appears to be 8 CPW resonators, but the experiment uses only 7 of them. What is the reason for that?

- A related question, in the MPO experiment, are the resonators measured directly through individual output ports (not shown in Fig 1g), or are they all probed via the buffer mode?

- Overall, I think that labeling Fig 1g with port functions and component IDs (eg “resonator 1”, “buffer”, “ATS”, etc.) will help a lot with orientation around the device and its function.

5. Attenuation of the delayed pulse. The experiment shows the delayed pulse attenuated, this is not surprising because of losses in the CPW resonators. However, I think the authors should say something about how that loss figures in the attenuation. Is the total attenuation limited by the loss rate of the worst resonator? Is it set by the sum of all resonator loss rate? Can the authors report resonator internal Q, not just kappas. Basically, the reader would want to know “how good would my resonators need to be to attempt a similar experiment?”.

6. Why not more/fewer resonators? What is special about 7 resonators, and why did the authors not use more or less resonators in the experiment? It would be informative to understand the trades here. Are there diminishing returns to using more resonators (perhaps less pulse distortion but maybe more loss, more pumps to tune up)? What is the minimum number of resonators that is acceptable, and what does it depend on?

Additional minor comments:

- Fig. 4(a) I assume the x-axis is a log scale? Please indicate that explicitly.

- P6 left, middle paragraph. Resonator frequency is indicated to 1 kHz precision. Do you really know this frequency that well? Does the frequency fluctuate at all? I would recommend relaxing the precision here to order of kappa, unless of course, the precision of the experiment is truly at the kHz level.

- P6 left, last paragraph: “The added noise is probably due to our strong parametric flux pump.” This doesn't make much sense as is. Are you referring to the strength of the coupling between the flux lines and the modes? Anything else?

- I think “MPO” needs to be defined before use

- In Methods: junction patterning. Are the junctions made using a Dolan bridge or some other technique?
- In supplement 1: paragraph 1 refers to an “ATS junction array”, I think this is not mentioned before. What array? is it used to make the ATS inductor? How many junctions, what is their critical current. What are the critical currents of the main ATS junctions that are shown in Fig.1?
- The experimental setup in supplement Fig.1 is not really referred to anywhere in the supplement, rather it is described in the methods section of the main paper. Consider moving the discussion to the supplement. It would also help if the connections to the device, as shown in the wiring diagram, could be tied to port labels on the device picture in Fig.1g of the main paper.

Reviewer #1 (Remarks to the Author):

The authors introduce the concept of a Parametrically Addressable Delay Line (PADL) as a parametrically programmable delay line. They demonstrate its main features and functionalities. Remarkably they show translating pulses in time and swapping of two pulses. I believe that such a PADL will be of great interest to the community working on the realization of quantum computing with superconducting circuits. I think that this work should be granted publication in Nature Communications, if the authors reasonably address my concerns, reported below.

We greatly appreciate your feedback on our submission. We agree that PADL will be of interest to the circuit QED community. In the following sections, we hope to address each of your concerns.

1. The first paragraph in the introduction should be expanded, and focus more on why delay lines can be instrumental for realizing quantum computing. There are only two references given and I would expect to see more. Also, references on the realization of a quantum computer based on superconducting circuits are missing.

We have added a longer discussion of how delay lines can be used to realize hardware-efficient quantum computing. We have also added references quantum computing using superconducting circuits. These edits can be found in the first paragraph of the paper.

2. I have some hard time understanding what is the real fidelity of the device. In the “fidelity and added noise” section, the authors first say that they measure fidelity of 95%, but when they then normalize $g(t)$ by the same factor as $f(t)$, the fidelity drops to 21%. So, what is the true device fidelity? My concern is then, would a device with such a low-fidelity be useful? Is there any threshold value that one should aim at achieving? How can one improve it, without using the ultra-high Q cavities mentioned by the authors? How much would the fidelity improve using state-of-the-art CPW resonators with higher Q?

Our goal is to differentiate whether the infidelity is due to pulse distortion or due to loss. Consider an input pulse with envelope $f(t)$ and delayed pulse with envelope $g(t)$. The input pulse should be normalized such that $|f(t)|^2$ integrates to one, corresponding to unit detection probability. This normalization accounts for the gain/attenuation in our measurement chain. If we also normalize $g(t)$ such that $|g(t)|^2$ integrates to one, then the overlap integral of $f(t)$ and $g(t)$ neglects any contributions to infidelity coming from the loss that $g(t)$ experiences in the PADL and directly compares the shape of $f(t)$ and $g(t)$. We find that this is equal to 0.95. However, if we normalize $g(t)$ by the same exact number that we normalized $f(t)$, the overlap integral takes into account the losses that $g(t)$ experiences in the PADL (such as from CPW loss). We find that this is equal to 0.21. Therefore, the true fidelity of this particular device is 0.21. With these losses, it is unlikely one can use error correction to realize a quantum computer. We have tried to clarify this on line 399.

The threshold value to achieve depends on the particular error correction scheme, as well as other rates (such as the rate at which flying qubits interact with the ancilla qubit in cluster state-based proposals). Examples of necessary delay line loss rates can be found in section VI of Ref. 2 in our submission. One simple approach to improving the fidelity would be to increase the bandwidth of the buffer mode. This would allow one to send much narrower pulses in time and therefore use shorter delay times relative to the fixed T_1 of the CPW resonators.

However, given that the fidelity is limited by CPW loss, ultimately very high fidelities will require very high Q cavities. State-of-the-art CPW resonators based on Tantalum can achieve $Q_i \sim 15e6$. We numerically simulated the equations of motion but considered our buffer mode to be lossless and our

CPW resonators to have $Q_i = 15e6$ and found $F = 0.991$ should be possible. We have included these simulations in Supplementary Fig. 5Se and included a discussion on line 198 of the Supplement.

3. In the conclusion, the authors suddenly talk about performing “process tomography on encoded qubits”. This comes out of the blue and no reference or any explanations are given. should put references when talking about performing

We agree that the mention of process tomography is abrupt and we have provided context on line 504.

I have also some minor comments:

4. The “Results” section starts with a subtitle reading “Implementing a parametric delay line with ATS”. Until this point there is no mention of what an ATS is in the text, so ATS should be removed from the subtitle.

We have removed “ATS” from the heading.

5. Acronym for FSR is not introduced to the readers

We have introduced the acronym on line 122 when free spectral range is first mentioned.

6. In the caption of Fig 2, the authors talk about ADC traces. Again, ADC needs to be defined somewhere if it is used. Also, if the traces are reported in terms of photon flux β_{in} and β_{out} , these quantity needs to be defined in the main text rather than in the supplementary information. Same comments apply to the caption of Figure 3. “ADC data” is also mentioned elsewhere in the text.

We clarified this and introduced “ADC” on line 296 of the main text. We have also defined photon flux and mean fields in the main text on line 320.

7. In the supplementary, Figures should be called with a different label. Maybe Fig #S, to distinguish them from figures in the main text Also the acronym ADC is used without any introduction.

We have updated the Figures to #S format in the supplement and updated the references to these figures accordingly in the main text and supplement. We also introduced “ADC” in the supplement on line 147.

Reviewer #2 (Remarks to the Author):

The manuscript describes a method for emulating a compact tunable delay line using parametric techniques. The chip uses 7 on-chip CPW cavities capacitively coupled to a buffer resonant mode that is modulated by a flux applied to an asymmetric tunable SQUID inductance. By applying appropriate parametric drives it is possible to control and manipulate microwave pulses that can reflect off of the buffer mode that is coupled to an input/output feedline. The Authors show the ability to delay the pulse over short times, delay for longer times by storing it in the resonators by turning off the parametric drives for storage time, rapidly stepwise advance or translate the phase, and finally, swap the order of pulses by reversing the evolution in time.

I find this work novel and a practical tool for use in future quantum information systems. These features

have not been achieved before in a very compact and convenient design that uses the same circuit resources already employed in existing superconducting circuits for quantum computing. I think this work will be of significance to the field and related fields and that these techniques will find use in helping develop future quantum information processors and communication channels. I believe that work presented supports the conclusions and claims, but I do have further comments and questions below that should be addressed. I believe there is enough detail provided in the methods for the work to be reproduced, but again see my comments below.

I think the work could be published in Nature Communications in principle after edits based on the comments below:

Thank you for your feedback. We agree that PADL will be useful in future quantum information systems. In the following sections, we hope to address each of your concerns.

1) Fig. 1: It'd be good to show a zoom in of the junctions in the ATS. At the scale shown, it's hard to tell what the JJs look like or how many there are.

We have added a zoomed in image of the ATS and its junctions in the Supplement.

2) Fig. 1: The cartoon pictures are also confusing to me. I believe the red pulse (same color as the buffer resonator?) is not delayed and the blue pulse (same color as the resonator modes that get parametrically coupled) is delayed, however they are close together in the graph with time and far apart on the cartoon of the waveguide? What are the two colors for, can you specify? Why is there two red and two blue pulses in the graphs? Is this meant to represent something at a beginning time and then at an ending time? It would be clearer to show one red and blue pulse delayed in one plot, then have another plot at a later time to show the delay between the two pulses with respect to each other at a later time -i.e., are they farther apart, closer together, or did they swap positions. The waveguides look clear but the blue and red pulses on them seem too far apart on these. Also, I think the little curved arrows are trying to say the pulse went into the delay line or came off it? Maybe describe this in the caption and/or label in and out?

Thank you for pointing out that Fig. 1 needs to be improved. We have updated Fig. 1 specifically to address your questions. The original red and blue pulses were not correlated with pulses stored in the buffer or CPW resonators, but you are correct that it was very unclear. We have recolored the pulses to orange and have differentiated input pulses and delayed pulses with dashed lines and solid lines, respectively. We also made the pulses a pair of pulses with different amplitudes to emphasize that the pulse-swapping experiment (Fig. 1e) swaps two input pulses, whereas the other experiments (Fig. 1b-d) do not. We also took your suggestion of explicitly labeling the input and output.

3) I see 8 CPW's coupled to the buffer, but only 7 modes listed in the paper?

Throughout these experiments, we only parametrically coupled 7 of the 8 fabricated CPWs. We had mentioned in a footnote that one of the fabricated CPWs was observed to have bad frequency noise, which is likely due to a two-level system (TLS) defect. We have moved this discussion from a footnote to the main text on line 276.

4) What are the ghost like streaks in Fig. 3a) at constant time at 2, 4, and 6 us, extending from $\tau = 0$ to 2 us, 2 to 4 us, and 4 to probably 6 us, respectively. Are they a concern for high fidelity operation?

Those streaks show the later photon echoes and are exactly the experimental signatures we are expecting to see. The bright streak at time = 2 us extending from $\tau = 0$ to 2 us is the stored pulse that gets delayed by 2 us, as shown by the black trace in Fig. 3b. When the PADL is continuously parametrically driven such that the parametric delay line has an FSR of 500 kHz, we expect the inputted pulse to be delayed by

time = $1/500$ kHz = 2 us. The dimmer streak at time = 4 us extending from $\tau = 2$ to 4 us is the stored pulse that gets delayed by 4 us, as shown by the blue trace in Fig. 3b. In this experiment, we turn off the parametric drives for $\tau < 2$ us. Therefore, the pulse that would have been delayed by 2 us cannot be re-emitted into the environment and is effectively delayed by another roundtrip. However, we turn on the parametric drives before $\tau = 4$ us, and therefore the pulse is re-emitted at time = $2/500$ kHz = 4 us. Similarly, the dimmest streak at time = 6 us corresponds to the case where we turn off the parametric drives for $\tau < 4$ us and so the inputted pulse is delayed by three roundtrips and is re-emitted at time = $3/500$ kHz = 6 us. This is shown by the red trace in Fig. 3b.

5) It seems that the Fidelity integral should be a function of τ . As seen in Fig. 3c) the amplitude of the shifted pulse decreases in amplitude with longer τ s. How will this effect performance and usefulness of the delay line? If the Fidelity is only maximum at T_{rt} , what can be done to improve performance over a larger range of τ s?

You are correct that the fidelity integral is a function of τ because the amplitude of $g(t)$ decays at longer τ (predominantly due to CPW loss). In the main text discussion section titled “Fidelity and added noise” we are specifically considering the case when the PADL is being continuously driven i.e. $\tau = 0$. We have clarified this point on line 386. When the parametric drives are being rapidly changed, fidelity is no longer maximized at T_{rt} . For example, fidelity is maximized at $2 \times T_{rt}$ and $3 \times T_{rt}$ in Fig. 3b and at $T_{rt} + \tau$ in Fig. 3d. To improve fidelity over a larger range of τ s, we need to improve the Qs of the CPW resonators.

6) In Fig. 3e), although the pulse appear reversed in time, the amplitudes suggest they were just flipped over. Would a simulation removing loss and reflections show improvement for this behavior?

We have included this time-domain simulation in Supplementary Fig. 6S. To emphasize that the two pulses are swapped (and not just experiencing an amplitude flip), we gave the two pulses different amplitudes. We see that when the detunings are swapped, the pulses are indeed swapped.

7) Shouldn't $\kappa_b = \kappa_{b,e} + \kappa_{b,i}$ be used in equation (1) of SM?

Yes, κ_b should be used in the denominator and $\kappa_{b,e}$ in the numerator. We have now also expressed explicitly that $\kappa_b = \kappa_{b,e} + \kappa_{b,i}$ (sum of extrinsic and intrinsic loss) on line 125 of the SM.

8) I assume $\kappa_k = \kappa_{k,e} + \kappa_{k,i}$, so how did you separate the vales for $\kappa_{k,e}$ & $\kappa_{k,i}$ for the fits in Table II?

You are correct that $\kappa_k = \kappa_{k,e} + \kappa_{k,i}$ is the total loss of the the k -th CPW resonator. We are able to separate intrinsic and extrinsic losses by directly probing the CPW resonators with a VNA (i.e. not through the buffer), which is possible due to parasitic capacitances between the CPW resonators and the readout transmission line. We have clarified this point on line 38 of the SM and line 460 of the main text.

9) Figure 5 in SM was not there. I looked at the Arxiv version Fig. 9.

We sincerely apologize for this confusion. We will double-check that the revised version contains Fig. 5 of the SM (which is indeed arxiv fig. 9).

10) The end of the paper would be improved by discussing what improvements must be made for practical use. SM Note 4 models the system and tries to improve the fidelity. Can these tests point to a direction for improvement? How can losses be improved in the CPW modes? Could they be lumped-

elements instead? How can the mode spacings be compensated if not equal (as stated in the intro) or made to be more equal to ensure performance? Could tweaks be made to operational frequencies and pulses to improve existing performance? How can the imperfections in impedance mismatches be solved? I think these questions and their answers must be included in the paper to show this device's viability.

Thank you for your suggestion. Our numerical simulations indicate that losses in the CPW modes are the dominant source of infidelity in our system. These losses can be improved by using state-of-the-art Tantalum CPW resonators. We have numerically simulated the case where CPW $Q_i = 15e6$ and found $F = 0.991$ should be possible, and we have included this simulation in Supplementary Fig. 5S. You are correct that the CPW modes could be replaced with lumped element modes; any resonator that can hybridize with the buffer lumped element mode should work.

Furthermore, as you suggested, tweaks in the pulses could be used to immediately improve performance. For example, by increasing the buffer bandwidth (i.e. by increasing the capacitive coupling between the buffer mode and the readout), one can use larger bandwidth pulses (i.e. narrower temporal FWHM). This would enable you to work with shorter delays such that the delay is even shorter relative to the CPW lifetime.

Impedance mismatches in the future can be addressed by integrating qubits with the PADL on the same chip. This would avoid any inevitable impedance mismatches that come from packaging your memory element (i.e. the PADL) and your source (i.e. the qubit).

The mode spacing between the CPW resonators actually does not need to be compensated because we can always tune the parametric drive frequencies to precisely place the parametrically converted CPW photons relative to the buffer frequency. This is one of the benefits of our device.

We have updated the final paragraph of our Discussion to reflect these improvements.

Reviewer #3 (Remarks to the Author):

The authors demonstrate a device, built with an asymmetrically threaded squid (ATS) parametrically connecting several superconducting CPW resonators, which the authors can operate as an effective delay line. Operation relies on the ATS mediating frequency conversion between an input mode (called the “buffer mode”) and the ensemble of resonators, with carefully tuned (detuning, phase, and amplitude) parametric pumps. The authors show microsecond-scale programmable delay of an input pulse, storage of pulses, and temporal swapping of pulse sequences. Programmability is achieved by dynamically controlling the parametric pump detunings. The authors additionally calibrate their device parameters, specifically the added noise, with separate experiments operating the device in amplifier/oscillator mode near threshold.

This is a really cool experiment with a novel device. While the application space is not very well articulated, I think this type of device (and indeed the device concept) could become an important tool in superconducting quantum information processing, perhaps in architectures based on cavity bosonic states. I therefore would support publication in Nature Communications, but I would like the authors to address some issues detailed below, mainly with the presentation of the work.

Thank you for your feedback and appreciating how cool our experiment is. We have tried to motivate how useful delay lines can be for quantum computing protocols in the introduction.

1. Analogy of delay line. I think the way that the analogy is introduced, and is used throughout the paper perhaps is going a bit too far to be useful. The introduction, in combination with the description associated with Fig. 1 (a-e), planted in my head a concept of the device that is actually the wrong one, and not until middle of page 4 I was able to actually understand what the experiment is actually doing and what the device topology is. My (hopefully now correct) understanding of the experiment is that an input pulse is spread by parametric conversion into a collection of resonant storage modes. Control of the detunings of the pumps can arrange it so that the pulse only interferes constructively back at the buffer I/O mode at particular times, hence it is effectively delayed. I would strongly recommend that the authors present a high level description of what's actually going on here (perhaps along the lines of my attempted explanation above), before driving the delay-line analogy. I also find Fig. 1 (a-d) not particularly helpful, and if the authors could come up with a more intuitive visualization that refrains from evoking a physical extended line or fiber topology for the experiment, that would improve the accessibility of the paper.

You are correct that our virtual delay line works by parametrically converting a data pulse into excitations in a collection of storage modes, and we agree that this is a very clear way of summarizing our experiment. We have emphasized this prominently on line 88 of the introduction, where we introduce and summarize our experiment. We also rewrote the very first paragraph of the Results section (starting on line 115 of the main text) to motivate your description and to de-emphasize the physical waveguide analogy. We also re-organized Fig. 1 to introduce the parametric processes underlying our device before introducing the waveguide analogy. We believe that with these edits, the waveguide analogy is clearer and still useful in describing our experiment.

2. Pump tuneup and setting the detunings. An important experimental bit of information is how to choose the correct detunings for all the pumps, and I find that information missing.

- What expression would we use to target these detunings, and how this is done procedurally in the lab.
- Are pump phases important? Do the pumps have to be coherent, and to what accuracy? What would be the effect of fixed phase offsets on the pumps? What would be the effect of relative phase drifts?

Thank you for your question. To be consistent with the paper, we would first like to clarify that we have one flux pump (given by $\epsilon_p(t)$ in Eq. 1 of the main text) and seven parametric drives (given by $\epsilon_k(t)$ in Eq. 2 of the main text) that off-resonantly drive the buffer. These frequencies are shown in Fig. 3S in the Supplement.

To engineer a parametric delay line with a desired detuning Δ_k (as in Eq. 4 of the main text), one first chooses a flux pump frequency ω_p . Then one chooses the drive frequencies ($\omega_{d,k}$) such that $\omega_{d,k} = \omega_p - (\omega_b + \Delta_k) + \omega_k$, as discussed in Eq. 3 of the main text. In principle, the flux pump frequency can be arbitrary because the parametric drive frequencies can always be chosen to satisfy Eq. 3 of the main text. In practice, as the parametric drive frequencies become more detuned from the buffer, they need to be stronger to achieve the same small displacement β_k on the buffer mode (see line 244 of the main text). We chose a flux pump frequency such that the parametric drive frequencies were roughly symmetrically placed around the buffer frequency, which made it such that we had enough RF power to generate all the drives from one AWG. We have clarified these points on line 256 of the main text.

The overall phase of the flux pump is not important because $\epsilon_p(t)$ scales the entire interaction Hamiltonian and thus constitutes a global phase. The phases of the parametric drives are very important; we explicitly modify these phases in the experiment shown in Fig. 3c and 3d to continuously translate the delayed pulse in time. Random relative phase shifts would cause the different parametric delay line modes

to not rephase, but these random phase shifts would have to be quite drastic. In the present work, all of our drive tones are provided from one channel of an AWG, so we do not worry about relative phase shifts. Even if future PADL experiments called for multiple signal sources providing the parametric drives, relative phase drift should not be a problem. These signal sources only need to remain coherent over the time of the experiment (a few microseconds in our case), and state-of-the-art signal sources have phase drifts that are many orders of magnitude slower than this.

3. Why ATS? A key component of the device is the ATS, which serves as part of the buffer mode, and is also providing the parametric coupling when pumped. I find the explanation of why they use an ATS not entirely convincing. The 3 reasons given on Page 2 would suggest that a DC SQUID could be used instead (3 wave mixing, inductive reactance, can use in MPO mode). Is there a specific property about the nonlinearity of the ATS that makes it advantageous in this application, over a flux pumped DC SQUID? When designing this device, we were worried about applying many off-resonant drives to a circuit element with a bounded potential (such as a DC SQUID). This also applies to using a so-called SNAIL to get 3-wave mixing. The inductor in the ATS makes the potential unbounded. One could also use a so-called RF-SQUID (same circuit geometry as a fluxonium), but for this geometry, we were concerned about anharmonicities. However, we did not attempt this experiment with these circuit elements. We have clarified this on line 204 of the main text.

4. Are there 7 or 8 CPW resonators?

- Fig. 1g shows what appears to be 8 CPW resonators, but the experiment uses only 7 of them. What is the reason for that?

- A related question, in the MPO experiment, are the resonators measured directly through individual output ports (not shown in Fig 1g), or are they all probed via the buffer mode?

- Overall, I think that labeling Fig 1g with port functions and component IDs (eg “resonator 1”, “buffer”, “ATS”, etc.) will help a lot with orientation around the device and its function.

Throughout these experiments, we only parametrically coupled 7 of the 8 fabricated CPWs. We had mentioned in a footnote that one of the fabricated CPWs was observed to have bad frequency noise, which is likely due to a two-level system (TLS) defect. We have moved this discussion from a footnote to the main text on line 276.

In the MPO experiment we directly readout the CPWs through the readout port (i.e. not through the buffer), which is possible due to weak parasitic capacitances. We have clarified this on line 460 of the main text.

We have also followed your suggestion and added labels to Fig. 1g, indicating the input/output line and labeled the modes by their frequencies (so as not to add too much text).

5. Attenuation of the delayed pulse. The experiment shows the delayed pulse attenuated, this is not surprising because of losses in the CPW resonators. However, I think the authors should say something about how that loss figures in the attenuation. Is the total attenuation limited by the loss rate of the worst resonator? Is it set by the sum of all resonator loss rate? Can the authors report resonator internal Q, not just kappas. Basically, the reader would want to know “how good would my resonators need to be to attempt a similar experiment?”.

The total attenuation of the delayed pulse is neither limited by the loss rate of the worst storage resonator nor the sum of all the resonators. Recall that the delayed pulse is parametrically converted into excitations living in a collection of storage resonators. Therefore, the frequency component of the input pulse at

Δ_k is approximately attenuated at a rate κ_k , so the total attenuation depends on the pulse's spectrum and how the storage resonators are parametrically arranged. We have clarified this on line 411 of the main text.

6. Why not more/fewer resonators? What is special about 7 resonators, and why did the authors not use more or less resonators in the experiment? It would be informative to understand the trades here. Are there diminishing returns to using more resonators (perhaps less pulse distortion but maybe more loss, more pumps to tune up)? What is the minimum number of resonators that is acceptable, and what does it depend on?

The figure of merit with delay lines is the delay-bandwidth product (DBP). For a simple overcoupled cavity, we have $DBP=1$ (the delay is $2\pi/\kappa$ and the bandwidth is $\kappa/2\pi$). However, for our parametric delay line, we have $DBP=N-1$ where N is the number of resonators that are being parametrically coupled to the buffer. Basically, the more resonators you parametrically couple, the longer delays you can achieve. As you pointed out, the tradeoff is the need to tune up more parametric drives (including crosstalk between the increasing number of drive tones), as well as the need to physically arrange more CPWs on your chip. Therefore, there is nothing special about 7 resonators, and this experiment can be done with more or fewer resonators.

One limit to our approach is when the buffer resonator capacitance is dominated by CPW coupling capacitances. We expect this to occur at around 20 CPW resonators. Beyond that number, we would need to redesign the storage resonator to have higher impedance. However, we have not explored these limits thoroughly at this point.

Understanding the minimum acceptable number of resonators depends on the desired experiment. Here is one way to approximate the minimum number of resonators for the simplest possible experiment: delaying a pulse by more than its temporal FWHM. The narrowest temporal pulse that one could use will have $t(\text{FWHM}) \sim 2\pi/kb$ where kb is the buffer bandwidth. Therefore, from the parametric DBP we have that $t(\text{delay})/t(\text{FWHM}) \sim N-1$. To visually differentiate the pulses we require $t(\text{delay}) > 2t(\text{FWHM})$ and thus one would require a bare minimum of 3 resonators.

Additional minor comments:

- Fig. 4(a) I assume the x-axis is a log scale? Please indicate that explicitly.

We have updated the figure caption to reflect this.

- P6 left, middle paragraph. Resonator frequency is indicated to 1 kHz precision. Do you really know this frequency that well? Does the frequency fluctuate at all? I would recommend relaxing the precision here to order of κ , unless of course, the precision of the experiment is truly at the kHz level.

We have updated Table I in the Supplement where we report our device parameters to include error bars, and we have clarified what the error bars mean in line 44 and line 49 of the Supplement. For the CPW resonator parameters, we repeatedly measured them over 81 samples and observed standard deviations of a few kHz. Therefore, we have left the reported values to kHz precision.

However, in the process of revisiting the data, we realized that we had reported our linewidths and parametric coupling strengths to precisions beyond the uncertainty. We have updated this in the supplementary tables. Thank you for bringing this to our attention.

- P6 left, last paragraph: “The added noise is probably due to our strong parametric flux pump.” This doesn't make much sense as is. Are you referring to the strength of the coupling between the flux lines and the modes? Anything else?

We are referring to potential heating effects from sending a strong RF signal onto our chip. For example, the PCB signal lines do not superconduct. We have clarified this on line 483 of the main text.

- I think “MPO” needs to be defined before use

We have defined MPO on line 107 of the main text.

- In Methods: junction patterning. Are the junctions made using a Dolan bridge or some other technique?

We have clarified the junction patterning on line 554 of the main text.

- In supplement 1: paragraph 1 refers to an “ATS junction array”, I think this is not mentioned before. What array? is it used to make the ATS inductor? How many junctions, what is their critical current. What are the critical currents of the main ATS junctions that are shown in Fig.1?

We have clarified that the ATS inductor is formed by a JJ array on line 183 of the main text. We have also included the number of junctions and the energy of junction array on line 17 and 21 of the supplement, and we have reported the junction energy of the SQUID junctions in Table 1S of the supplement.

- The experimental setup in supplement Fig.1 is not really referred to anywhere in the supplement, rather it is described in the methods section of the main paper. Consider moving the discussion to the supplement. It would also help if the connections to the device, as shown in the wiring diagram, could be tied to port labels on the device picture in Fig.1g of the main paper.

We have moved the discussion of the experimental setup to the supplement. We have also updated the wiring diagram so it is tied to port labels on the device.

REVIEWERS' COMMENTS

Reviewer #1 (Remarks to the Author):

I strongly support the manuscript for publication. With the last version, the authors have addressed all my concerns and comments.

Reviewer #2 (Remarks to the Author):

I feel the authors have addressed all my comments and have made the appropriate changes. This work should be published in its revised form.

Reviewer #3 (Remarks to the Author):

The authors addressed all my comments, and I think the paper is now acceptable for publication in Nature Communications after a minor revision. I would like to see the authors improve the discussion on the following points:

- 1) The discussion in the authors' rebuttal to my question (6), regarding the minimum number of resonators is very informative, and I would encourage the authors to discuss this in the paper, perhaps in the supplement.
- 2) Please make a clearer distinction in the manuscript between the flux pump and the parametric drives. They are fed to the chip from different generators and via different ports. Please also comment on their relative amplitudes - are "drives" much weaker than the flux pump? How does the amplitude of stored pulse compare to those of the drives and pump?
- 3) Please report the range of the resonators internal Q 's in the main text (or refer to the supplement table 1S). This could be for example on page 4 after line 269

Reviewer #1 (Remarks to the Author):

I strongly support the manuscript for publication. With the last version, the authors have addressed all my concerns and comments.

We are happy that we addressed your concerns and comments.

Reviewer #2 (Remarks to the Author):

I feel the authors have addressed all my comments and have made the appropriate changes. This work should be published in its revised form.

We are happy that we addressed your comments with appropriate changes.

Reviewer #3 (Remarks to the Author):

The authors addressed all my comments, and I think the paper is now acceptable for publication in Nature Communications after a minor revision. I would like to see the authors improve the discussion on the following points:

We are happy that we addressed your major comments. Below, we hope to address your remaining minor comments.

1) The discussion in the authors' rebuttal to my question (6), regarding the minimum number of resonators is very informative, and I would encourage the authors to discuss this in the paper, perhaps in the supplement.

We have included our discussion of the delay-bandwidth product and the minimum number of resonators in Supplementary Note 5.

2) Please make a clearer distinction in the manuscript between the flux pump and the parametric drives. They are fed to the chip from different generators and via different ports. Please also comment on their relative amplitudes - are "drives" much weaker than the flux pump? How does the amplitude of stored pulse compare to those of the drives and pump?

We have included the on-chip powers for the parametric drives, the flux pump, and the store pulses in Supplementary Note 2. We have also emphasized the difference between the parametric flux pump (magnetic flux threading the ATS loops) and the parametric drives (off-resonant drives that are capacitively coupled to the buffer mode) in the first paragraph of the section titled "Parametric control of stored wavepackets."

3) Please report the range of the resonators internal Q 's in the main text (or refer to the supplement table 1S). This could be for example on page 4 after line 269.

We have referenced the resonator Q_i values in the main text in the last paragraph of the section titled "Implementing a parametric delay line."